# RBF-Based Camera Model Based on a Ray Constraint to Compensate for Refraction Error

**DOI:** 10.3390/s23208430

**Published:** 2023-10-12

**Authors:** Jaehyun Kim, Chanyoung Kim, Seongwook Yoon, Taehyeon Choi, Sanghoon Sull

**Affiliations:** School of Electrical Engineering, Korea University, Seoul 02841, Republic of Korea; jhkim@mpeg.korea.ac.kr (J.K.); cykim@mpeg.korea.ac.kr (C.K.); swyoon@mpeg.korea.ac.kr (S.Y.); taehyeon@korea.ac.kr (T.C.)

**Keywords:** camera calibration, camera behind transparent shield, refraction, radial basis function (RBF) approximation

## Abstract

A camera equipped with a transparent shield can be modeled using the pinhole camera model and residual error vectors defined by the difference between the estimated ray from the pinhole camera model and the actual three-dimensional (3D) point. To calculate the residual error vectors, we employ sparse calibration data consisting of 3D points and their corresponding 2D points on the image. However, the observation noise and sparsity of the 3D calibration points pose challenges in determining the residual error vectors. To address this, we first fit Gaussian Process Regression (GPR) operating robustly against data noise to the observed residual error vectors from the sparse calibration data to obtain dense residual error vectors. Subsequently, to improve performance in unobserved areas due to data sparsity, we use an additional constraint; the 3D points on the estimated ray should be projected to one 2D image point, called the ray constraint. Finally, we optimize the radial basis function (RBF)-based regression model to reduce the residual error vector differences with GPR at the predetermined dense set of 3D points while reflecting the ray constraint. The proposed RBF-based camera model reduces the error of the estimated rays by 6% on average and the reprojection error by 26% on average.

## 1. Introduction

Cameras are extensively employed across a variety of industries [1,2], highlighting the crucial role of camera calibration in rectifying image distortions. To comprehend the relationship between the actual three-dimensional (3D) ray and the two-dimensional (2D) point represented in the image, reliance on camera models is imperative. These models estimate two types of projections: forward projection, deducing the 2D point on the image plane corresponding to the observed 3D point or ray, and backward projection, calculating the ray originating from the derived 2D point. Simplified camera models, such as the pinhole and fisheye models, are widely adopted for these projections. Nevertheless, practical applications frequently demand the incorporation of transparent shields for safeguarding camera lenses against external threats. In these scenarios, the simplistic camera models prove to be inadequate, as they fail to accurately represent the refracted ray.

To address this issue, one method involves estimating light refraction by taking into account the shape of the transparent shield [3,4,5,6]. This method seeks to determine the alteration of light paths as they traverse through the shield, thereby facilitating a more accurate representation of the refracted ray, as depicted on the left side of Figure 1. Subsequently, the refracted ray outside the shield is computed utilizing Snell’s Law [7]. Nonetheless, these methodologies necessitate detailed information pertaining to the shield, posing challenges in their application when alterations to the shield occur.

To overcome these limitations, an alternative approach involves directly estimating the relationship between the 2D point within the image and the ray outside the transparent shield [8,9,10,11,12,13,14]. This method represents the relationship between the 2D point and the ray using a more complex parameterized model, the so-called generalized camera model. The parameters of this model are then fitted using calibration data, i.e., the 2D points in the image and their corresponding rays (or points in 3D space). This approach does not depend on information about the transparent shield, allowing for recalibration and parameter adjustment even when minor changes occur to the shield.

Following the introduction of the generalized camera model by Grossberg et al. [8], research shifted towards gathering data to refine these models. Early efforts by Ramalingam et al. [10] and Rosebrock et al. [15] used 2D patterns to establish corresponding rays for each pixel. Given the sparse nature of 2D pattern corners in 3D space, direct interpolation or spline-based methods were employed.

Recent advancements in 3D observation tools have emphasized the use of 3D calibration data for improving ray representation in intricate camera systems. Miraldo et al. [11] used a Vicon sensor for 3D data and applied Plücker coordinates for ray representation. Yet, their approach faced accuracy issues under increasing noise due to inadequate constraints on the Plücker coordinates. In tackling the issue of windshield refraction, Verbiest et al. [13] presented a model that was more resilient to noise, using splines to represent refraction on the 2D image plane. Nonetheless, their presumption that refraction is linearly proportional to inverse depth has been shown to have certain constraints [14]. Recognizing these constraints, Choi et al. [14] presented a residual camera model that addresses errors by compensating within the 3D space. Nonetheless, as model parameters expand, the demand for additional calibration data rises, leaving the issues of data sparsity and measurement noise as persistent challenges.

To address these challenges, we adopt an approach similar to Choi et al. [14] but aim to enhance it. The overall model is divided into two parts: the pinhole camera and the other component that compensates for refraction errors. Initially, we fit the parameters of the pinhole camera to the refraction-distorted calibration data as much as possible. Subsequently, we estimate and correct the incurred errors in the form of 3D vectors, called residual error vectors. However, unlike the existing method that interpolates sparse data, we opt for an RBF-based regression model and perform parameter optimization, considering data noise and sparsity.

This model is refined utilizing two kinds of objective functions. Initially, Gaussian Process Regression (GPR) is employed as a reference to formulate the posterior distribution of residual error vectors stemming from the calibration data. This strategy amplifies the model’s robustness against data noise. Nonetheless, its performance is still lacking in unobserved 3D regions where 3D observations are absent. To address the residual error vectors in such regions, we introduce a ray constraint.

The ray constraint we propose stems from the interplay between backward and forward projection. Specifically, when we derive two 3D points in a well-observed region from the backward projection of a single 2D point on the image plane, the ray connecting these points should also correct the residual error vector for points in the unobserved region to align with the original 2D point via forward projection. This technique utilizes a ray from the well-observed region to correct the residual error vector in unobserved areas, thereby enhancing the accuracy of the forward projections in less observed territories. As a result, the proposed model surpasses existing methods and GPR, especially in environments with sparse or noisy data. Furthermore, the model benefits from rapid processing, owing to its non-iterative nature like the previous method [14].

We demonstrate the superiority of the proposed RBF-based camera model through comparative experiments with various levels of noise and data density in the calibration data. The distance between the estimated ray and ground-truth 3D point is reduced by an average of 6%. Furthermore, a substantial reduction of approximately 26% occurs in the reprojection error of the sparsely calibrated existing model.

In conclusion, the innovative regression-based camera model offers a robust solution to the challenges introduced by transparent shields in camera calibration. By overcoming the limitations of conventional models and integrating an additional restriction, the proposed method ensures precise camera calibration.

The contributions of this research are summarized as follows:We introduce an RBF-based camera model designed to counteract the refraction error engendered by diverse types of transparent shields.Through optimizing the RBF kernel utilizing the outcomes of GPR and ray constraint, the model attains robust performance, even amidst 3D data sparsity and measurement noise present in the calibration data.The efficacy of the proposed method is evidenced by a reduction in the distance between the estimated ray and ground truth 3D point and a decrease in reprojection error by approximately 6% and 26%, respectively.

The remainder of this paper is organized as follows. Section 2 provides an overview of related work. Next, Section 3 discusses the proposed RBF-based camera model in detail. Then, Section 4 presents the experimental results, and Section 5 presents limitations and future work. Finally, Section 6 concludes the paper.

The list of abbreviations is at the end of the paper, in Table 1.

## 2. Related Work

This section examines existing camera models suitable for representing complex camera systems, such as those equipped with transparent shields or reflective objects. These models can be broadly classified into three categories: a simple camera model, a complex camera model that directly models the ray path, and a generalized camera model that establishes a direct mapping from a 2D image point to a ray without requiring ray path estimation. An overview of the related work can be found in Table 2.

### 2.1. Simple Camera Models

The pinhole camera model [16] and fisheye camera model [17] are widely employed in most commercial cameras. Typically, these models are calibrated using 2D calibration objects [16,18], although they can also be calibrated using 1D (line) calibration objects [19], or even without calibration objects at all [20,21], due to their simplicity.

However, when these camera models are applied to refracted mediums, Kang et al. [3] conducted experiments to demonstrate the feasibility of the simple camera model. They showed the effectiveness of the pinhole camera model when incorporating distortion parameters. They assumed flat transparent shields positioned perpendicular to the camera’s direction. Therefore, when the transparent shield has a more complex shape or is not placed perpendicular to the camera direction, more complex camera models may be required.

### 2.2. Camera Model Explicitly Models the Ray Path

Different camera models have been developed to address specific scenarios involving the path of the physical ray in complex camera systems. First, for catadioptric camera systems that include both a lens and mirror, Geyer et al. [22] introduced a unifying model capable of adapting to various mirror types placed in front of the camera. They explicitly parameterized the reflection from the mirror.

**Table 2 sensors-23-08430-t002:** Brief explanations and pros and cons for related works.

Method		Explanation
Camera model that explicitly models the path of the camera ray	Cassidy et al. [5]	Propose a multi-view stereo system for reconstructing objects in a transparent medium.
		Pros: Effective 3D reconstruction of objects in the transparent medium.
		Cons: Requires precise information about the transparent medium.
	Pavel et al. [4]	Model the surface of the transparent shield using RBF kernel.
		Pros: Can handle the slightly deformed surface of the transparent shield.
		Cons: Requires information about the transparent shield.
	Yoon et al. [6]	Propose model and the calibration method for the partially known curved-shape transparent shield.
		Pros: Applicable to partially known transparent shields.
		Cons: Requires additional observation points inside the shield.
Generalized camera model	Grossberg et al. [8]	Firstly propose a concept of generalized camera model.
		Pros: Applicable to any camera system.
		Cons: Requires an active 2D calibration pattern.
	Ramalingam et al. [10]	Propose a calibration method that relies on a minimum of three 2D calibration patterns.
		Pros: Requires only 2D calibration patterns.
		Cons: Exhibits lower performance due to the use of 2D calibration data.
	RoseBrock et al. [15,23]	Employ splines for interpolating the 2D calibration pattern.
		Pros: Effectively interpolates the sparse 2D calibration data.
		Cons: Exhibits lower performance due to the use of 2D calibration data.
	Beck et al. [12]	Use spline to model the refracted ray.
		Pros: Efficiently models refracted rays caused by transparent shields.
		Cons: Assumes the refracted rays converge on one point, which is not valid.
	Miraldo et al. [11]	Directly estimate the ray from RBF using 3D calibration data.
		Pros: Estimates the ray accurately under low-noise conditions.
		Cons: Exhibits reduced performance as noise levels increase.
	Verbiest et al. [13]	Estimate the windshield refraction using 2D residual error vectors.
		Pros: Effectively models the refraction in the windshield.
		Cons: Uses the strong assumption that the refraction is inversely proportional to the depth.
	Choi et al. [14]	Estimate the error of simple camera model using 3D error vectors.
		Pros: Demonstrates high performance and robustness against data noise in various types of camera systems.
		Cons: Requires dense 3D calibration data and can have mismatches between forward and backward projection models.

In cases where the camera is positioned behind a refractive object, Cassidy et al. [5] explicitly modeled the refractive medium and ray path, applying a refraction model for multiview stereo. Additionally, Pavel et al. [4] addressed scenarios with cylinder-shaped transparent shields, using radial basis functions (RBFs) to explicitly model the uneven shield surface. Yoon et al. [6] tackled the modeling of stereo camera systems positioned behind curved transparent shields. They parameterized the curved shield and optimized the shape parameters, while also utilizing additional feature points on the inner surface of the shield to effectively estimate its shape.

These methods require knowledge of the shape of the mirror or the transparent object. Our goal is to model refraction without relying on prior knowledge of the shield shape. Therefore, we have chosen to adopt the approach using the generalized camera model.

### 2.3. Generalized Camera Model

Generalized camera models, as introduced by Grossberg et al. [8] and Pless et al. [9], establish a direct mapping from the image to the camera ray. Unlike other models, they avoid assuming global properties of rays in the image plane, such as radial symmetry, and instead rely on local smoothness assumptions. These models have significantly more parameters compared to pinhole or fisheye camera models, making calibration challenging.

Calibration of these models typically involves observing at least two 3D points corresponding to one image point. The ray passing through these two 3D points represents the ray corresponding to that image point. To achieve dense observations for all pixels in the image plane, some methods utilize active calibration patterns [8]. In contrast, others replace the need for active patterns by interpolating the rays or corners of 2D calibration patterns [15,23].

Ramalingam et al. [10] introduced a calibration method for the generalized camera model designed to work across various camera systems. Additionally, Schops et al. [24] proposed dense and accurate 2D calibration patterns and a corresponding calibration method tailored for the generalized camera model. They initiated the parameters for the ray using Ramalingam et al.’s calibration method [10], which solved collinearity equations for observed 3D collinear points. Subsequently, they employed an iterative optimization process to fine-tune the ray and minimize errors, similar to the approach taken by RoseBrock et al. [15,23].

Furthermore, Beck et al. [12] applied the generalized camera model to a camera positioned behind a windshield. However, their assumption that rays from outside the shield converge at a single point is not valid for real-world scenarios.

These methods rely on 2D calibration patterns. However, as camera systems become more complex, 2D calibration patterns may prove insufficient to capture the camera’s characteristics. Consequently, some approaches turn to 3D calibration patterns [11,25]. In the case of Miraldo et al. [11], they utilized an RBF-based regression model and incorporated 3D points obtained from a Vicon sensor along the data containing both 3D points and their corresponding 2D image points for calibration purposes. They represented rays using Plücker coordinates. Nevertheless, the constraint of Plücker coordinates is not reflected to the cost of the parameter optimization, and the method becomes sensitive to noise, as discussed by Choi et al. [14].

Verbiest et al. [13] explicitly modeled the windshield, approximating it as a spherical shape for simplified ray path modeling. They employed this spherical shield model solely for the estimation of intrinsic and extrinsic camera parameters. Subsequently, they applied a generalized camera model, utilizing splines to compensate for the error introduced by refraction. It is important to note that this approach assumes a linear correlation between the distortion caused by the transparent shield’s refraction and the inverse depth. However, this assumption may not hold true in all scenarios, as highlighted in recent research by Choi et al. [14].

Choi et al. [14] introduced a novel approach known as the residual camera model. This model is designed to estimate and compensate for errors that arise from simple camera models, such as the pinhole model [16] or the fisheye camera model [17]. In scenarios where these simple camera models provide satisfactory performance, the residual camera model proves effective due to its ability to handle noise robustly and accurately capture errors using dense 3D observations.

Their method is versatile and applicable to various situations, including cameras positioned behind transparent shields and fisheye or catadioptric cameras with misaligned lenses or mirrors. This versatility stems from the fact that simple camera models can perform adequately in these scenarios.

However, their method relies on the availability of dense 3D points to accurately capture errors in the simple camera model. Furthermore, their noniterative forward and backward projection process can lead to inconsistencies between these processes. To address these issues, we propose an RBF-based camera model that leverages the noise-robustness of Gaussian Process Regression (GPR) and introduces a novel constraint about the ray. This approach aims to mitigate the challenges posed by sparse observations and the mismatch between the forward and backward processes, particularly when dealing with cameras positioned behind transparent shields.

## 3. Methods

This section introduces the regression-based camera model in the following sequence: residual camera model, residual error vector regression, and optimization. The entire process of calibration of the proposed camera model is illustrated in Figure 2, and the inference process is presented in Figure 3.

### 3.1. Residual Camera Model

The residual camera model [14] comprises the backbone model and residual components that compensate for the error of the backbone model. The backbone model employs simple camera models, such as the pinhole or fisheye camera model [17]. In this case, we use the pinhole camera model as a backbone model. The residual components are interpreted as error vectors corresponding to the 2D or 3D space. Building on the previous work that found that interpreting error vectors in 3D space outperforms their interpretation in 2D space [14], this study adopts this approach, and a brief overview is provided.

For the 2D image point, the 3D ray is estimated using the pinhole camera model and residual model. First, for the 3D point w and corresponding 2D image point x, the pinhole camera model is described as follows:(1)$${\hat{x}}_{b}=\pi \left(w\right),$$
(2)$$\begin{array}{ccc} l(x)=\{s+\gamma d|s & = & t,\\ d & = & \pi^{-1}\left(x\right)-t,\\ \gamma & \in & R\}, \end{array}$$
where l(x) represents the ray of the pinhole camera model from the 2D image point x, π is the forward projection of the 3D point, and $\pi^{-1}\left(\cdot \right)$ denotes the inverse process of the projection. In addition, t is the position of the camera center. We used the pinhole camera model as a backbone model; thus, π(w) is calculated using the equation $K\left(R^{T}\left(w-t\right)\right)$, where K is an intrinsic parameter of the pinhole camera model, and R,t are extrinsic parameters. Moreover, $\pi^{-1}\left(x\right)$ is $RK^{-1}\tilde{x}+t$, where $\tilde{x}$ is a homogeneous coordinate of the 2D image point. We assume that the initial noisy values of K, R, and t are known. The parameters are updated to approximate the refraction using these as initial values, as presented in the first row in Figure 2. For the calibration data $x={\left\{\left(x_{i},w_{i}\right)\right\}}_{i=1}^{N}$, which are the 2D image point and corresponding 3D point, K, R, and t are updated iteratively to minimize the reprojection error for the calibration data [16].

After attaining the parameters of the pinhole camera model, the forward and backward projections of the residual model in Figure 3 are described as follows:(3)$$\begin{array}{cc} \hat{x}= & \pi (w+r^{\left(f\right)}\left(w\right)), \end{array}$$
(4)$$\begin{array}{ccc} \hat{l}\left(x\right)=\left\{s+\gamma d\right|s & = & w^{\left(1\right)}+r^{\left(b\right)}\left(w^{\left(1\right)}\right),\\ d & = & w^{\left(2\right)}+r^{\left(b\right)}\left(w^{\left(2\right)}\right)-s,\\ \gamma & \in & R\}, \end{array}$$
where $\hat{l}\left(x\right)$ is the ray of the residual camera model, $w^{\left(1\right)}$ and $w^{\left(2\right)}$ represent points selected on l(x) that are closer to and farther from the camera, respectively, and $r^{\left(f\right)}\left(\cdot \right)$ and $r^{\left(b\right)}\left(\cdot \right)$ are the residual error vectors corresponding to the 3D point. These complement the errors observed in forward and backward projections of the pinhole camera model, respectively. The residual error vectors for the calibration data ${\left\{\left(x_{n},w_{n}\right)\right\}}_{n=1}^{N}$ in the second row in Figure 2 are calculated as follows:(5)$$\begin{array}{c} r_{o}^{\left(f\right)}\left(w_{n}\right)=w_{n}-proj\left(w_{n},l\left(x_{n}\right)\right), \end{array}$$(6)$$\begin{array}{c} r_{o}^{\left(b\right)}\left(w_{n}\right)=proj\left(w_{n},l\left(x_{n}\right)\right)-w_{n}, \end{array}$$
where $proj(x_{n},l\left(x_{n}\right))$ is the projection of 3D point $x_{n}$ to the ray $l(x_{n})$.

Based on the *N* observed residual error vectors obtained in this manner, a method is required to estimate the residual error vector for any other arbitrary 3D point. For instance, in Choi et al. [14], the researchers obtained entire residual error vectors by smoothing and interpolating the observed residual error vectors.

This residual camera model [14] is simple and generally applicable, providing straightforward forward and backward projections. Furthermore, the model comprises noniterative operations, making it efficient for fast execution. Nevertheless, drawbacks remain in smoothing and interpolating the observed residual error vectors. First, the error vectors are determined through interpolation of observed vectors in the calibration data; thus, they are susceptible to cases of inadequate or sparse distribution within the calibration data. Second, when backward projecting to obtain a ray for an arbitrary 2D point and subsequently forward projecting it, it does not return precisely to the original 2D point. This outcome signifies accumulated errors when alternating forward and backward projections, posing challenges in such tasks as simultaneous localization and mapping.

### 3.2. Residual Error Vector Regression

To overcome the limitations of the existing methods, we first interpreted the sparse residual error vector from the probabilistic perspective. The distributions of residual error vectors are described as follows, considering the observation noise in $r_{o}^{\left(f\right)}$ and $r_{o}^{\left(b\right)}$:(7)$$\begin{array}{c} r^{\left(f\right)}\left(w_{n}\right)∼N\left(r_{o}^{\left(f\right)}\left(w_{n}\right),\sigma_{n}^{2}I\right), \end{array}$$(8)$$\begin{array}{c} r^{\left(b\right)}\left(w_{n}\right)∼N\left(r_{o}^{\left(b\right)}\left(w_{n}\right),\sigma_{n}^{{}^{\prime }2}I\right), \end{array}$$
where N(·,·) is a Gaussian distribution, and $\sigma_{n}$ and $\sigma_{n}^{\prime }$ denote the standard derivations. Based on this perspective, we can determine the residual error vector for arbitrary 3D points using GPR. For instance, the forward residual error vector for an arbitrary point $w^{\prime }$ is calculated as follows: (9)$$\begin{array}{c} r_{G}^{\left(f\right)}\left(w^{\prime }\right)=K\left(w^{\prime },W\right){\left(K\left(W,W\right)+\sigma_{n}^{2}I\right)}^{-1}R, \end{array}$$(10)$$\begin{array}{c} R={\left[\begin{array}{ccc} proj\left(w_{i},l\left(x_{i}\right)\right)-w_{i}, & \cdots , & proj\left(w_{N},l\left(x_{N}\right)\right)-w_{N}) \end{array}\right]}^{T}, \end{array}$$
where $w^{\prime }$ denotes an arbitrary 3D point, and $K(w^{\prime },W)$ signifies the similarity between $w^{\prime }$ and the set of 3D points in calibration data W. The backward residual error vectors are obtained similarly. Thus, we can estimate the residual error vector for any arbitrary point $w^{\prime }$.

This approach demonstrates robust performance when the residual error vector is locally smooth and approximately centered. However, GPR remains susceptible to imbalances in the distribution of calibration data because it relies heavily on nearby data points for inference. For instance, the model may produce highly inaccurate results at locations where calibration data are absent.

To enhance performance in such a scenario, we employ another parameterized regression model which allows for the straightforward incorporation of the additional constraint that can handle the problem of the absence of the data, presented in the last row of Figure 2 and Section 3.3.2.

We present a regression model based on RBFs to obtain improved residual error vectors. This model stores vectors and confidences for control points scattered throughout the considered 3D region. Given a 3D point w, it references the vectors of neighboring control points to calculate the residual error vector. The RBF is a function for computing similarity or distance between data points. It is employed to determine the similarity between the control point and w. Furthermore, as in [26], we add a first-order polynomial term for $w={\left[w^{\left(x\right)},w^{\left(y\right)},w^{\left(z\right)}\right]}^{T}$. Through this RBF-based regression model, we can compute the forward residual error vector along the *x*-axis, as follows:(11)$$\begin{array}{ccc} r_{R}^{\left(f,x\right)}\left(w\right) & = & \sum_{i}^{M}\phi_{\sigma }^{\left(f,x\right)}\left(w,c_{i}^{\left(f\right)}\right)\lambda_{i}^{\left(f,x\right)}+w^{T}{\left[a_{x}^{\left(f,x\right)},a_{y}^{\left(f,x\right)},a_{z}^{\left(f,x\right)}\right]}^{T}+a_{0}^{\left(f,x\right)}, \end{array}$$(12)$$\begin{array}{ccc} \phi_{\sigma }^{\left(f,x\right)}\left(w,c_{i}^{\left(f\right)}\right) & = & \sqrt{∥w-c_{i}^{\left(f\right)}{∥}_{2}^{2}+{\left(\sigma_{\left(f,x\right)}\right)}^{2}}, \end{array}$$
where $\phi_{\sigma }^{\left(f,x\right)}\left(w,c_{i}^{\left(f\right)}\right)$ denotes the RBF kernel to obtain the similarity between the 3D point w and control point $c_{i}^{\left(f\right)}$. In addition, ${\left\{\lambda_{i}^{\left(f,x\right)}\right\}}_{i=1}^{M}$ represents the weights associated with control points, $a_{x}^{\left(f,x\right)},a_{y}^{\left(f,x\right)},a_{z}^{\left(f,x\right)},a_{0}^{\left(f,x\right)}$ represent the coefficients for the linear polynomial, and $\sigma_{\left(f,x\right)}$ is a hyperparameter which is set before obtaining other parameters. Using the values obtained for the *x*-, *y*-, and *z*-axes, we calculate the forward residual error vector as follows: $r_{R}^{\left(f\right)}\left(w\right)={\left[r_{R}^{\left(f,x\right)}\left(w\right),r_{R}^{\left(f,y\right)}\left(w\right),r_{R}^{\left(f,z\right)}\left(w\right)\right]}^{T}$. The backward residual error vector is similarly computed using the same approach. The method for determining the parameters ${\left\{\lambda_{i}^{\left(f,x\right)}\right\}}_{i=1}^{M}$, $a_{x}^{\left(f,x\right)}$, $a_{y}^{\left(f,x\right)}$, and $a_{z}^{\left(f,x\right)}$ for this purpose is described in Section 3.3.

The forward and backward projection for a 2D point x and a 3D point w according to the proposed RBF-based camera model is calculated by substituting $r^{\left(f\right)},r^{\left(b\right)}$ to $r_{R}^{\left(f\right)},r_{R}^{\left(b\right)}$ in Equations (Equation 3) and (Equation 4):(13)$$\begin{array}{c} \hat{x}=\pi \left(w+r_{R}^{\left(f\right)}\left(w\right)\right), \end{array}$$
(14)$$\begin{array}{cc} \hat{l}\left(x\right)=\left\{s+\gamma d\right| & s=w^{\left(1\right)}+r_{R}^{\left(b\right)}\left(w^{\left(1\right)}\right),\\ \; & d=w^{\left(2\right)}+r_{R}^{\left(b\right)}\left(w^{\left(2\right)}\right)-s,\\ \; & \gamma \in R\}.\\ \; & w^{\left(1\right)},w^{\left(2\right)}\in l\left(x\right). \end{array}$$

### 3.3. Optimization of the RBF Regression Model

In this subsection, we present the process of optimization of the proposed RBF regression model. We show the reference objective, ray constraint, and the combined objective for the RBF regression model.

#### 3.3.1. Reference Objective

The reference objective prevents the model from deviating from GPR. To fit the GPR, we used cross-validation to obtain additional parameters, such as the type of the kernel and kernel parameters. However, overfitting occurs in cross-validation when the amount of observation data is low and when the observation noise is too low. To manage this problem, we obtained an additional small set of residual error vectors using interpolation from the observed residual error vectors. These residual error vectors are additionally used for fitting the GPR. We applied this for all experiments regardless of the noise and sparsity of the data.

After fitting GPR to the observed data, the reference objective is computed as the difference between the residual error vectors estimated using GPR ${\left\{r_{G}\left(\tilde{w}\right)\right\}}_{i=1}^{L}$ and those estimated using the RBF for the 3D points selected at regular intervals ${\left\{\tilde{w}\right\}}_{i=1}^{L}$, as follows:(15)$$\begin{array}{c} L_{Ref}^{\left(f,x\right)}=\frac{1}{L}\sum_{i}^{L}{\left(r_{R}^{\left(f,x\right)}\left({\tilde{w}}_{i}\right)-r_{G}^{\left(f,x\right)}\left({\tilde{w}}_{i}\right)\right)}^{2}, \end{array}$$
where $L_{Ref}^{\left(f,x\right)}$ represents a reference objective for the *x*-axis component of the forward projection model, and $r_{G}^{\left(f,x\right)}\left({\tilde{w}}_{i}\right)$ denotes the *x*-axis component of $r_{G}^{\left(f\right)}\left({\tilde{w}}_{i}\right)$ in Equation (Equation 9). The objective is applied for each axis value and forward and backward model. For each forward and backward projection model, $L_{Ref}$ represents the sum of the cost for each axis. The 3D points are listed on the left side in Figure 4. From these points, we select control points using the *k*-means algorithm. The nearest 3D point ${\tilde{w}}_{i}$ from each centroid of the cluster is used for the control point. The control points are presented in Figure 4. After selecting control points, we optimize the RBF coefficient in Equation (Equation 17). We utilize the grid search for parameter σ in kernel $\phi_{\sigma }^{\left(f,x\right)}\left(w,c_{i}^{\left(f\right)}\right)$, checking the errors in Equation (Equation 15).

For the forward projection model, we used an additional optimization step with another objective. By fixing the control points and σ, we updated the RBF coefficient to minimize the sum of $L_{Ref}$ and the additional objective $L_{Ray}$ described in the following section.

#### 3.3.2. Ray Constraint

The fundamental idea embedded in the ray constraint is that all points on a ray correspond to a single 2D point on the image plane. In other words, all the points belonging to that ray should return to their original 2D point through the camera model’s forward projection. This motivation is valid when the forward projection model is correct. However, as listed in the last row in Figure 2 (the points ${\tilde{w}}_{1}$ in the unobserved region where 3D calibration data do not exist), the inaccurate forward residual error vector leads to the incorrect forward-projected point. To remedy this, we apply a constraint to the forward residual error vector, ensuring that 3D points on the estimated ray are projected onto the same image point.

This constraint requires an accurate ray. Therefore, we estimate a ray using only the backward residual error vectors in the well-observed region where the 3D calibration data are observed. With the obtained precise ray, we apply the ray constraint to the forward residual error vector for all regions traversed by the ray. In addition, to ensure the unobserved region resides in the well-observed region, we obtained calibration data from the near and far points from the camera along the *z*-axis. This ensures the accuracy of the ray and establishes an effective ray constraint for the area between closer and farther points along the *z*-axis.

In Figure 5, to realize the ray constraint, we first obtain a ray $l(x_{j})$ induced from a 2D point $x_{j}$ using the pinhole camera model. The backward projection model operates by correcting two 3D points on the ray from the pinhole camera model. The sampled two points are obtained from the well-observed region near and far from the camera. The corrected ray is accurate because it is obtained using backward residual error vectors in the well-observed region. After obtaining *J* rays ${\hat{l}}_{j}\left(x_{j}\right)$, we uniformly sample 3D points ${\left\{{\bar{w}}_{p}\right\}}_{p=1}^{P}$ from these rays. Then, we apply the forward residual error vectors to these points. If the forward residual error vector is correct, then the modified points ${\bar{w}}_{p}+r_{R}^{\left(f\right)}\left(\bar{w_{j}}\right)$ should be on the pinhole ray to ensure the forward projection of $\bar{w_{j}}$ is the same as the initial 2D point $x_{j}$. The ray constraint is formulated as follows:(16)$$\begin{array}{c} \begin{array}{cc} L_{Ray}^{\left(x\right)}=\; & \frac{1}{P}\sum_{j}^{J}\sum_{{\bar{w}}_{p}\in P_{j}}{\left({\bar{w}}_{p}^{\left(x\right)}+r_{R}^{\left(f,x\right)}\left({\bar{w}}_{p}\right)-proj^{\left(x\right)}\left({\bar{w}}_{p}+r_{G}^{\left(f\right)}\left({\bar{w}}_{p}\right),\hat{l}\left(x_{j}\right)\right)\right)}^{2},\\ \mathrm{where}\; & P_{j}={\left\{{\bar{w}}^{\left(1\right)}+\gamma_{i}\left({\bar{w}}^{\left(2\right)}-{\bar{w}}^{\left(1\right)}\right)\right\}}_{i=1}^{K},\;{\bar{w}}^{\left(1\right)},{\bar{w}}^{\left(2\right)}\in \hat{l}\left(x_{j}\right),\\ \; & P=\sum_{j}^{J}n(P_{j}),\;\mathrm{and}\;{\bar{w}}_{p}=\left[{\bar{w}}_{p}^{\left(x\right)},{\bar{w}}_{p}^{\left(y\right)},{\bar{w}}_{p}^{\left(z\right)}\right]. \end{array} \end{array}$$
Here, *J* denotes the number of sampled rays, and *P* represents the number of sampled points for all sampled rays. We obtain the rays by applying the backward residual model obtained using GPR. We apply a backward projection model for each image point uniformly sampled in the image plane. The above ray constraint is only for the *x*-axis component of the residual error vector. The objective is calculated for the rest of the axis. Additionally, $proj^{\left(x\right)}\left(w,l\left(\cdot \right)\right)$ indicates the *x*-axis component of a point from the projection of w to the 3D ray l(·). For the forward projection model, $L_{Ray}$ represents the sum of the cost for each axis.

#### 3.3.3. Combined Objective

In summary, the objectives for the regression models for the forward and backward residual error vectors are $L_{Ref}$ and $L_{Ref}+\alpha_{Ray}L_{Ray}$, respectively, where $\alpha_{Ray}$ is the weight for the ray constraint. We minimize the above objectives separately for each axis, which involves solving a weighted least square solution for a matrix equation to obtain the optimal solution. For instance, the equation for the *x*-axis of the forward residual error vector is expressed as follows:(17)$$\begin{array}{c} \left[\begin{array}{ccccc} \phi_{\sigma }^{\left(f,x\right)}\left({\tilde{w}}_{1},c_{1}^{\left(f\right)}\right) & \cdots & \phi_{\sigma }^{\left(f,x\right)}\left({\tilde{w}}_{1},c_{M}^{\left(f\right)}\right) & {\tilde{w}}_{1}^{T} & 1\\ \vdots & \ddots & \vdots & \vdots & \vdots \\ \phi_{\sigma }^{\left(f,x\right)}\left({\tilde{w}}_{L},c_{1}^{\left(f\right)}\right) & \cdots & \phi_{\sigma }^{\left(f,x\right)}\left({\tilde{w}}_{L},c_{M}^{\left(f\right)}\right) & {\tilde{w}}_{L}^{T} & 1\\ \phi_{\sigma }^{\left(f,x\right)}\left({\bar{w}}_{1},c_{1}^{\left(f\right)}\right) & \cdots & \phi_{\sigma }^{\left(f,x\right)}\left({\bar{w}}_{1},c_{M}^{\left(f\right)}\right) & {\bar{w}}_{1}^{T} & 1\\ \vdots & \ddots & \vdots & \vdots & \vdots \\ \phi_{\sigma }^{\left(f,x\right)}\left(w_{P},c_{1}^{\left(f\right)}\right) & \cdots & \phi_{\sigma }^{\left(f,x\right)}\left({\bar{w}}_{P},c_{M}^{\left(f\right)}\right) & {\bar{w}}_{P}^{T} & 1 \end{array}\right]\left[\begin{array}{c} \lambda_{1}^{\left(f,x\right)}\\ \vdots \\ \lambda_{M}^{\left(f,x\right)}\\ a_{x}^{\left(f,x\right)}\\ a_{y}^{\left(f,x\right)}\\ a_{z}^{\left(f,x\right)}\\ a_{0}^{\left(f,x\right)} \end{array}\right]=\left[\begin{array}{c} r_{G}^{\left(f,x\right)}\left({\tilde{w}}_{1}\right)\\ \vdots \\ r_{G}^{\left(f,x\right)}\left({\tilde{w}}_{L}\right)\\ proj^{\left(x\right)}\left({\bar{w}}_{1}+r_{G}^{\left(f\right)}\left(w_{1}\right),{\hat{l}}_{1}\right)-{\bar{w}}_{1}\\ \vdots \\ proj^{\left(x\right)}\left({\bar{w}}_{P}+r_{G}^{\left(f\right)}\left(w_{P}\right),{\hat{l}}_{J}\right)-{\bar{w}}_{P} \end{array}\right], \end{array}$$
where $proj^{\left(x\right)}\left(\cdot ,\cdot \right)$ is the *x*-axis component for proj(·,·), ${\left\{{\tilde{w}}_{i}\right\}}_{i=1}^{L}$ denote the points for $L_{Ref}$ in Equation (Equation 15), and $w_{i,j}$, ${\bar{w}}_{i}$, and ${\hat{l}}_{j}$ represent the notation in Equation (Equation 16). The weights are assigned as 1 for the first *L* rows and $\alpha_{Ray}$ for the remaining rows. In Figure 6, we assess the proposed residual error vectors for the forward model compared to those from Choi et al. [14].

## 4. Experiments

This section presents the experimental results of the RBF-based camera model. We first detail the dataset and evaluation metrics. Then, we compare the model with the existing camera models. Finally, we discuss the ablation study for the dependency of the proposed camera model for the parameters and data configuration, as well as assessing its computational load.

### 4.1. Dataset

We utilized three types of shields in our experiments; the first and second types are plane and curved shields, referred to as ’plane’ and ’sphere’, respectively, both with a uniform thickness of 5 mm. The third type, known as the ’dirty plane’, is a planar shield with uneven thickness, designed to simulate a shield with a deformed outer surface. Figure 7 provides visual representations of these transparent shields used in our experiments.

For generating test 3D points, we considered distances ranging from 1 to 10 m along the *z*-axis (parallel to the camera direction). Notably, the 3D calibration points were sparser compared to the test 3D points across all axes. Specifically, for the *z*-axis, which aligns with the camera’s viewing direction, we only used calibration data at 1 and 9 m, as shown in Figure 8. This choice was made to simulate a scenario with limited calibration data, as these data points yielded better results than alternative configurations, such as using 3D points at 5 and 9 m or 1 and 5 m. A detailed comparison of these results is presented in Section 4.7.4.

We employed MATLAB [27] to simulate our data. The calibration data consist of pairs, each containing a 3D point and its corresponding 2D image point. To simulate the observation noise, we added Gaussian noise to both the image and 3D points. The term ’noise level’ in our results denotes the standard deviation of this Gaussian noise. For instance, a noise level of 0.1 corresponds to a standard deviation of 0.1 pixels for the image and 0.1 cm for the 3D point, respectively.

### 4.2. Implementation Details

When performing backward projection, we selected two points along the ray derived from the pinhole camera model, as outlined in Equation (Equation 4). These points were chosen based on the minimum and maximum *z*-axis components of the 3D calibration points, represented as $z_{min}$ and $z_{max}$, respectively. Specifically, we defined the near and far points as $z_{min}$ + 20 cm and $z_{max}$ – 20 cm, respectively, along the ray originating from the pinhole camera model. This selection was made because the residual error vector can be more accurately estimated within the range of 3D calibration points as opposed to points near the boundaries of this range.

### 4.3. Evaluation Metric

In our evaluation process, we assessed the reprojection error during forward projection and measured the distance between the estimated ray and the ground truth (GT) 3D point in the context of backward projection. Additionally, we quantified the difference between the forward and backward models by examining the difference as defined in Equation (Equation 16). To provide a comprehensive view of our results, we employed both box plots and mean values (represented by green triangles across all experiments) since we conducted multiple iterations. For each scenario, we conducted a total of eight experiments. This was achieved by randomly introducing noise four times, and considering two different shield shapes for each shield type, as illustrated in Figure 7.

### 4.4. Results along the Noise Levels

We first examine the results of the proposed RBF-based camera model in scenarios with sparse data, where observations are only available at 1 and 9 m. In Figure 9, we present the results for noise levels ranging from 0.2 to 0.8. In the figure, ’RBF w. $L_{Ref}$’ denotes the proposed RBF-based camera model optimized exclusively with the reference objective, $L_{Ref}$, as defined in Equation (Equation 15). ’RBF w. $L_{Ref}$ and $L_{Ray}$’ is used for the proposed camera model optimized using both objectives.

In Figure 9, the first row displays the reprojection error results for each camera model. Notably, the proposed RBF-based camera model consistently demonstrates superior performance compared to all other methods. The presence of the $L_{Ref}$ component has a minimal impact on a plane-shaped shield, as the results of ‘RBF w. $L_{Ref}$’ closely resemble those reported by Choi et al. [14].

However, as the complexity of the shield shape increases, the influence of $L_{Ref}$ becomes larger, resulting in reduced error compared to the results from Choi et al. [14]. In contrast, ‘$L_{Ray}$’ proves to be effective for all shield types, indicating that the proposed ray constraint, $L_{Ray}$, is consistently effective regardless of shield shape and noise levels.

It is important to note that as noise levels increase, particularly in the case of the ‘Dirty plane’ shield, the performance gap in reprojection error between ‘$L_{Ref}$’ and ‘$L_{Ray}$ and $L_{Ref}$’ narrows. This is due to the amplified noise, which leads to increased errors in the backward ray, as illustrated in the last row of the figure.

The results from Verbiest et al. [13] have a lower error than the pinhole camera model in the ‘Dirty plane’ shield for reprojection in the first row and the ray error in the last row. This outcome is because the spline used in Verbiest et al. [13] is more flexible than the pinhole camera model. The spline can be fitted to the observation without any assumptions for the image plane globally, such as radial symmetric projection in the pinhole camera model. However, in the simpler shield type, Verbiest et al. [13] found larger errors than the pinhole camera model. This result is due to the disadvantage of using the spline, which is vulnerable to noise. This tendency can be observed when changing the noise level. When the noise increases, the performance gap between the pinhole and the model by Verbiest et al. [13] increases accordingly. Thus, when the shield shape is simple, the disadvantage of using the spline is much more prominent than the advantages.

The results obtained by Verbiest et al. [13] exhibit lower errors compared to the pinhole camera model when dealing with the ‘Dirty plane’ shield, both in terms of reprojection (first row) and ray error (last row). This discrepancy arises because the spline utilized by Verbiest et al. [13] offers greater flexibility than the pinhole camera model. The spline can adapt to observations without imposing global assumptions on the image plane, such as the radial symmetric projection inherent in the pinhole camera model. However, in the case of simpler shield types, Verbiest et al. [13] observed larger errors compared to the pinhole camera model. This outcome can be attributed to the drawback of using splines, which are more susceptible to noise than simple camera models.

In Figure 9, the second row illustrates the difference between forward and backward projection, which is calculated using Equation (Equation 16). The pinhole camera model and the model proposed by Verbiest et al. [13] show no difference between their forward and backward models. This is because the pinhole camera model employs the inverse matrix of forward projection, and Verbiest et al. [13] use an iterative process for backward projection.

Conversely, Choi et al.’s model [14] exhibits a substantial difference between forward and backward projections due to its non-iterative nature for both. In contrast, ‘RBF w. $L_{Ref}$’ showcases a lower difference between the forward and backward projections compared to the Choi et al. [14] model. This is attributed to the lower model error observed in both forward and backward projections, as indicated in the first and third rows. When ‘$L_{Ray}$’ is applied, the difference between the forward and backward projections decreases significantly. Furthermore, the degradation in performance with increasing noise is much smaller compared to the Choi et al. [14] model or ‘RBF w. $L_{Ref}$’.

### 4.5. Results with Low Noise Levels

We excluded the results obtained by Miraldo et al. [11] from Figure 9 due to significant errors occurring when the noise level equals or exceeds 0.2. Instead, we present a comparison with the Miraldo et al. [11] model in Figure 10, focusing on lower noise levels of 0, 0.05, 0.1, and 0.2.

In the first row of the figure, we observe the reprojection error. At lower noise levels, the Miraldo et al. [11] model outperforms other methods, especially for the ‘Dirty plane’ shield. However, as the noise level increases, the error escalates significantly. This trend is also reflected in the third row, which examines the distance between the estimated ray and the ground-truth 3D point. The increase in noise disrupts the optimization objective used by the Miraldo et al. [11] model, rendering it less representative of the actual distance between the ray and the observed 3D point, as pointed out by Choi et al. [14].

With the exception of the Miraldo et al. [11] model, when considering the reprojection error, the ‘RBF w. $L_{Ref}$’ exhibits a larger error compared to the Choi et al. [14] model when noise levels are low. This occurs because at low noise levels, interpolation tends to be more accurate in creating dense vectors. However, ‘RBF w. $L_{Ref}$ and $L_{Ray}$’ consistently demonstrate significantly lower reprojection errors than the Choi et al. [14] model, irrespective of the noise level.

Regarding the error between forward and backward projection, ‘RBF w. $L_{Ref}$’ shows a greater error compared to the Choi et al. [14] model, particularly for the ‘Dirty plane’ shield. Nevertheless, this error diminishes when ‘$L_{Ray}$’ is applied. The overall trend in the remaining results aligns with those observed in Figure 9.

### 4.6. Results When the Data Are More Sparse along the *x*-, *y*-Axis

When evaluating the effectiveness of the proposed method in scenarios with increased sparsity along the *x*- and *y*-axes for calibration data, we present the results in Figure 11 and Figure 12. In each figure, the *x*-axis represents the degree of sparsity along the *x*- and *y*-axes within the 3D space. Starting from the entire dataset, we progressively omit data to achieve 50%, 33%, and 25% of the remaining data points.

The calibration 3D points are visualized in Figure 13. In Figure 11, a relatively low noise level of 0.1 is used, resulting in minor errors. In such cases, for simpler shield shapes, the error of the proposed camera model aligns closely with that of Miraldo et al. [11]. However, with the complex ’Dirty plane’ shield, the proposed RBF-based camera model exhibits superior performance compared to Miraldo et al. [11]. Additionally, as the sparsity along the *x*- and *y*-axes increases, the performance gap between the proposed camera model and Miraldo et al. [11] widens.

In Figure 12, we present the results for a higher noise level. In terms of reprojection error and forward–backward projection, the proposed ray constraint consistently demonstrates its effectiveness, even as the data sparsity along the *x*- and *y*-axes increases. However, when evaluating the distance between the estimated ray and the ground-truth 3D point, we observe that for the simple shield shape, the proposed RBF-based camera model does not yield significant performance improvements compared to Choi et al. [14].

### 4.7. Ablation Study

This section conducts an ablation study of the proposed camera model, investigating its sensitivity to various parameters and the reference model. Furthermore, we compare results based on the sparsity of the calibration data and data configurations. Finally, we assess the computational load of the proposed camera model.

#### 4.7.1. Results According to the Weight for the Ray Constraint in Section 3.3.3

In Section 3.3.3, we introduced the ray constraint, $L_{Ray}$, and combined it with $L_{Ref}$, controlled by the weight parameter $\alpha_{Ray}$. The performance dependence on $\alpha_{Ray}$ is illustrated in Figure 14. The results indicate that the optimal value for $\alpha_{Ray}$ lies within the range of $1\times 10^{4}$ to $1\times 10^{5}$, which is notably high. This suggests that the ray constraint holds greater importance than the reference objective. However, when $\alpha_{Ray}$ exceeds $1\times 10^{5}$, the results deteriorate. This indicates that $L_{Ref}$ must remain close to the objectives for the initial observations of the residual error vectors. Therefore, we have chosen to set $\alpha_{Ray}$ to $1\times 10^{4}$ for the remainder of our experiments.

In terms of reprojection error, the gap between $\alpha_{Ray}=1$ and $\alpha_{Ray}=1\times 10^{4}$ widens, particularly for the shield of the dirty plane. Additionally, the error between the forward and backward projection models becomes larger with complex shield shapes. In the case of complex transparent shields, the forward residual error vectors require more observations or additional constraints to effectively model the errors introduced by these intricate shield shapes.

#### 4.7.2. Results of Changing the Reference Model

We explored the effectiveness of $L_{Ray}$ when changing the reference model to that of Choi et al. [14], as depicted in Figure 15. In this figure, ‘RBF w. $L_{Ref-interp}$’ signifies the RBF regression model optimized solely using $L_{Ref}$ as the reference model. However, it is important to note that the reference model is not GPR; rather, it is based on Choi et al.’s approach [14], which involves interpolation to densify sparse error vectors. ‘RBF w. $L_{Ref-interp}$ and $L_{Ray}$’ represents the model that incorporates the additional objective $L_{Ray}$.

In the first row, we observe an improvement in reprojection error when using interpolation as the reference model. However, when dealing with complex shield shapes like the ‘Dirty plane’ shield, employing GPR as the reference model yields superior performance compared to using interpolation.

The second row, similar to reprojection error, demonstrates that $L_{Ray}$ is effective in reducing the error between forward and backward projections, regardless of the reference model. Using GPR as a reference model consistently narrows the gap between forward and backward models for all shield types compared to using interpolation. In the third row, we observe the effectiveness of GPR, particularly for the ’Dirty plane’ shield.

#### 4.7.3. Results According to the Data Sparsity along the *z*-Axis

We conducted a comparison of results with varying data sparsity along the *z*-axis, as depicted in Figure 16. The *x*-axis in each figure represents the level of data sparsity along the *z*-axis. Moving from left to right in each figure along the *x*-axis, the calibration data are altered, ranging from 1 to 9 m, then down to observations at 1, 5, and 9 m, and finally at 1 and 9 m only.

As the calibration data become sparser, all camera models exhibit a performance decline. However, this degradation is reduced when using $L_{Ref}$ or both $L_{Ref}$ and $L_{Ray}$.

When dealing with a more complex shield shape and sparse calibration data, the performance gap between the proposed RBF-based camera model and Choi et al.’s model [14] becomes larger. This underscores the effectiveness of the proposed objectives, $L_{Ref}$ and $L_{Ray}$, in compensating for the challenges posed by sparse observations, particularly when dealing with complex shield shapes.

In the second row, which evaluates the error between forward and backward projections, the proposed camera model utilizing $L_{Ray}$ consistently exhibits significantly lower error than other methods. Moreover, the performance degradation is considerably small.

#### 4.7.4. Results According to the Configuration of Calibration Data

We analyze the results concerning the configuration of calibration data, as presented in Figure 17. Different colors represent various configurations of the calibration data. With the exception of the plane-shaped shield, the observations at 1 and 9 m consistently yield superior results compared to the configurations of 1 and 5 m and 5 and 9 m. This phenomenon can be attributed to the 1 and 9 m observation configuration offering a broader operational range.

When observing at 1 and 9 m, the model can interpolate the residual error vectors across the entire range from 1 to 9 m. However, with the 1 and 5 m or 5 and 9 m configurations, interpolation is limited to the respective subranges. Therefore, the calibration data at 1 and 9 m provide the most extensive operational range for the camera model.

In the case of the plane-shaped shield, while the error between forward and backward projections, as well as the error of the backward ray, is minimized when the model utilizes observations at 5 and 9 m only, the performance gap with the model using 1 and 9 m observations is much smaller than in other cases, suggesting similar results.

#### 4.7.5. Computational Load of the Proposed RBF-Based Camera Model

The computation time of the proposed method is presented in Table 3. The table displays both the mean and standard deviation of the computation time across eight trials, involving a total of 3948 test points. In this table, Choi et al. [14] exhibit faster computation than the proposed method, primarily because they employ interpolation, which is simpler than the RBF-based regression model. However, the computation complexity using big O notation is the same for both methods as O(n).

In contrast, Miraldo et al. [11] and Verbiest et al. [13] necessitate iterative procedures for either the forward or backward projections, resulting in significantly higher computational complexity than O(n). As a consequence, the difference in computational speed between Choi et al. [14] and the proposed method is relatively small when compared to the computational overhead introduced by Miraldo et al. [11] and Verbiest et al. [13].

## 5. Limitations and Future Work

The proposed RBF-based camera model shows effectiveness in various types of transparent shields. The proposed reference objective effectively captures the error of the pinhole camera model in the well-observed region and the ray constraint effectively complements the sparsity of the data. The proposed camera model mainly yields remarkable results when the data are sparse, and the outer shield surface is deformed. When the outer surface is deformed, the error of the pinhole camera model varies globally according to the 3D position. In this case, the residual error vector of the pinhole camera model in the unobserved region is difficult to estimate, making the ray constraint necessary.

Indeed, it is crucial to recognize the limitations of the proposed camera model. First, while it reduces errors between forward and backward projection processes, some errors persist and can be accumulated over repeated forward and backward projections. One way to address this is by implementing a hard constraint that ensures the error between forward and backward projections is reduced to zero.

Secondly, the effectiveness of the ray constraint hinges on the accuracy of the estimated ray. If the accuracy of ray estimation decreases, the impact of the ray constraint diminishes as well. To address this issue, incorporating an additional constraint for the estimated ray can be considered. This constraint would aim to maintain ray accuracy even in scenarios with sparse data or substantial observation noise.

In the future, the application of the proposed camera model can be extended to various types of camera systems, including those equipped with transparent shields, fisheye lenses, or reflective objects. Additionally, our current research focused exclusively on monocular camera calibration. Future studies could explore stereo camera calibration employing a residual camera model. In such a scenario, the observed residual error vector could be defined as the difference between the estimated stereo 3D points and the observed 3D points.

## 6. Conclusions

We have introduced an RBF-based camera model that incorporates a ray constraint, designed to address refraction errors in various types of transparent shields. This camera model utilizes RBF regression to estimate the residual error vector of the pinhole camera model, which serves as compensation for the error of the pinhole camera model. The optimization of this RBF regression model involved two key objectives, enhancing its ability to estimate pinhole camera model errors effectively.

The first objective focuses on minimizing the difference with the outcome produced by the GPR. This objective significantly boosts the model’s robustness, especially when dealing with noisy data. The second objective centers on the ray constraint, which ensures that 3D points on the estimated ray are projected onto a single image point. This constraint proves to be highly effective, particularly in situations where calibration data may be sparse or lacking density.

The proposed camera model effectively diminishes the reprojection error, the discrepancy between the estimated ray and the actual 3D point, and the differences between the forward and backward projection models. It demonstrates its efficacy across various types of transparent shields and showcases robustness to observational noise and data sparsity.

However, there are some limitations of the proposed camera model. First, the proposed camera model reduces errors between the forward and backward models, but there are residual errors that need further refinement. Second, the effectiveness of the proposed ray constraint relies on the accuracy of the estimated ray.

In the future, these limitations can be addressed by implementing enhanced constraints for residual error vectors. Additionally, while our proposed camera model is tailored for monocular camera systems operating behind transparent shields, future work could focus on extending its applicability to various monocular or stereo camera systems equipped with fisheye lenses or mirrors.

## Figures and Tables

**Figure 1 sensors-23-08430-f001:**
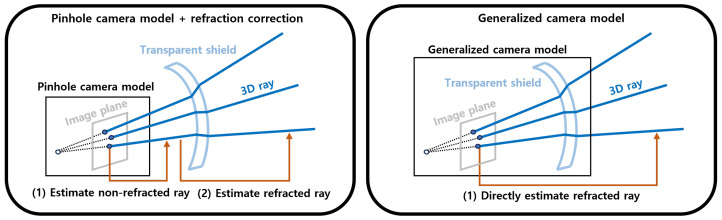
Two models depict a camera behind a transparent shield. The left model utilizes tracing of the precise ray path through the shield but needs the shield information. The right model, a generalized camera model, simply maps from the 2D point to the ray without detailing the ray path. We adopt the right model, especially when the shield shape is unknown.

**Figure 2 sensors-23-08430-f002:**
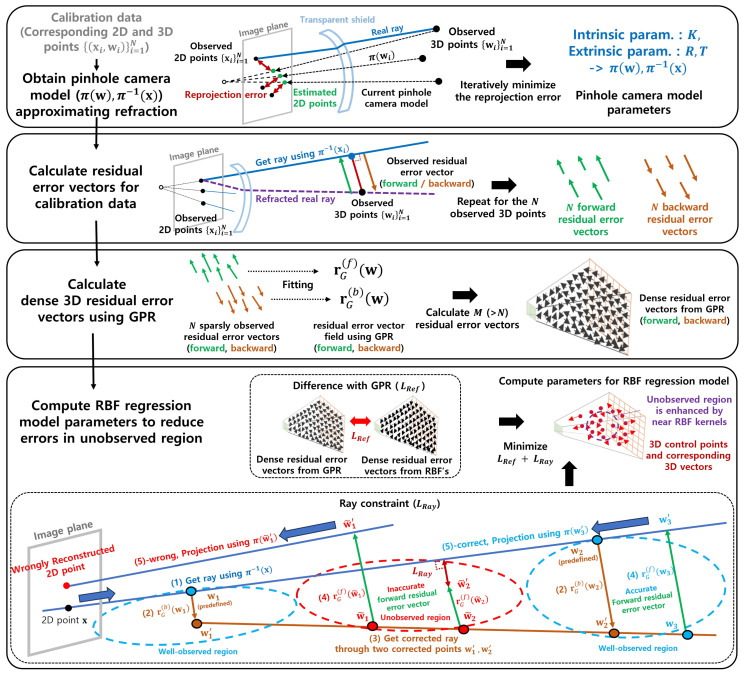
Calibration procedure for the proposed camera model. We determined the parameters of the pinhole camera model to reduce the reprojection error on the image plane for the calibration data. Next, we calculated the 3D sparse residual error vectors observed when applying the calibration data to the pinhole camera model. Then, we fit Gaussian Process Regression (GPR) to the sparse residual error vectors to obtain dense residual error vectors. Finally, to improve performance in unobserved regions due to 3D data sparsity, we propose an additional objective for residual error vectors, $L_{Ray}$ in the last row. Following the index in the figure, we provide one example of the ray constraint. (1) We obtain a ray from one image point x using the pinhole camera model $\pi^{-1}\left(x\right)$. Then, we select two points $w_{1}$ and $w_{2}$ from the observed near and far regions from the camera. (2) We correct $w_{1}$ and $w_{2}$ to $w_{1}^{\prime }$ and $w_{2}^{\prime }$ using backward residual error vectors, which are accurate because the error vectors are in a well-observed region. (3) We obtain the ray through the corrected points $w_{1}^{\prime }$ and $w_{2}^{\prime }$, which is a backward projected ray of the proposed camera model. If the forward residual error vector is correct, then the 3D points on the modified ray are projected again to the same 2D point x. (4) To perform the forward projection, we apply the forward residual error vector to the sampled points on the corrected ray. (5)-correct, When the forward residual error vector is correct because it is obtained from the well-observed region, the projected 2D point is the same as the original 2D point x. (5)-wrong, When the forward residual error vector is wrong because it is obtained from the unobserved region, the reconstructed 2D point is not the same as the original 2D point x. To create accurate forward residual error vectors in the unobserved region, we calculated the inconsistency that measures the error of forward residual error vectors $L_{Ray}$ in the figure. Finally, the RBF-based regression model is optimized by two objectives. The first is the objective to reduce the error vector difference with GPR, and the second is the ray constraint $L_{Ray}$.

**Figure 3 sensors-23-08430-f003:**
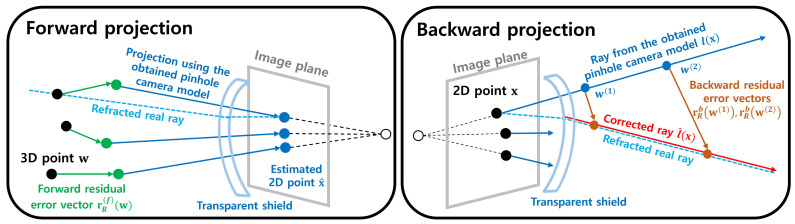
Forward and backward projections according to the proposed RBF-based camera model. The left side displays a forward projection, where the forward projection model estimates the projected point for the 3D points. The right side presents a backward projection where it outputs an estimated ray for 2D points. The forward projection applies the residual error vector for the input 3D point to correct it. The pinhole camera model performs forward projection. In backward projection, the input 2D point is first back-projected to the ray from the pinhole camera model. Then, the two points on the ray are selected, one near and one far from the camera. Then, the backward residual error vectors are applied to these points to obtain two modified 3D points. Finally, the ray through the two modified 3D points is the estimated 3D ray. For both projections, the residual error vector is obtained from the regression model.

**Figure 4 sensors-23-08430-f004:**
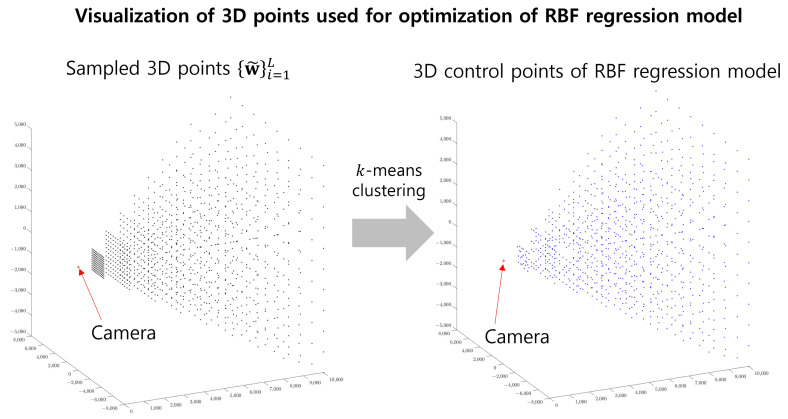
(**Left**) Sampled 3D points ${\left\{{\tilde{w}}_{i}\right\}}_{i}^{L}$ in Equation (Equation 15). The position was chosen with a uniform interval for each *z*-axis. (**Right**) Control points of the RBF regression model obtained from the sampled 3D points in the left figure. The nearest point for each centroid of the *k*-means algorithm was used for the control point, where *k* is 80% of the sampled 3D points.

**Figure 5 sensors-23-08430-f005:**
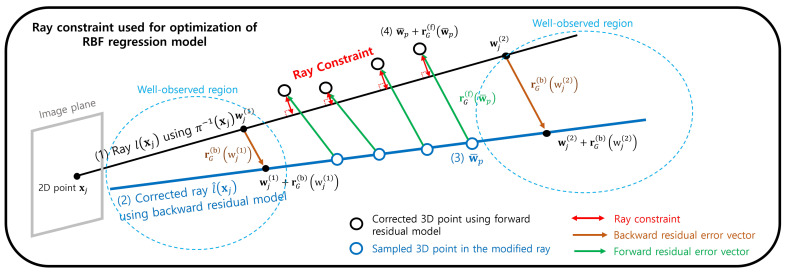
Ray constraint: Following the residual camera model, we corrected the $j^{th}$ ray from the pinhole camera model $l(x_{j})$ to the corrected ray $\hat{l}\left(x_{j}\right)$ with two sampled points in the ray from the pinhole camera model. These sampled points are obtained from the well-observed region where the residual error vectors are accurate. Then, we sampled 3D points ${\bar{w}}_{p}$ on the corrected ray and applied the forward residual to obtain the corrected points ${\bar{w}}_{p}$ and $r_{G}^{\left(f\right)}\left({\bar{w}}_{p}\right)$. The ray constraint is the forward residual error vector $r_{G}^{\left(f\right)}\left({\bar{w}}_{p}\right)$, which should be on the ray from the pinhole camera model $l(x_{j})$.

**Figure 6 sensors-23-08430-f006:**
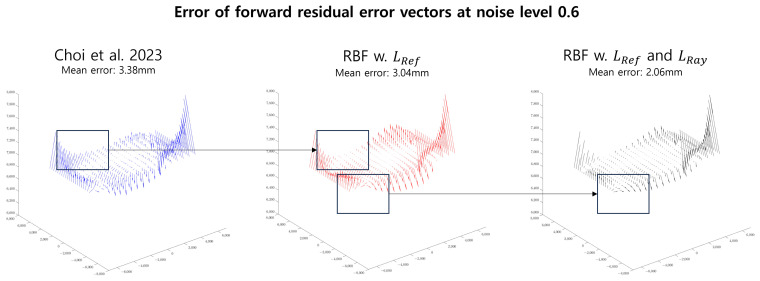
Visualization of the residual error vectors in the proposed RBF-based camera model. We checked the forward residual error vectors by determining the difference in the GT residual error vector 7 m from the camera along the *z*-axis when the calibration data only exist at 1 and 9 m. Thus, the accuracy of the proposed residual error vector at 7 m is much better than in the other methods. The left figure, ’Choi et al 2023’, presents the residual error vectors from Choi et al. [14], which have a large error. When we used the RBF regression model with $L_{Ref}$ only, which aims to approximate the result of the Gaussian Process Regression, the residual error vectors have a lower error than that found by Choi et al. [14], as depicted in the center. After applying $L_{Ray}$ (the ray constraint), the error is further reduced.

**Figure 7 sensors-23-08430-f007:**
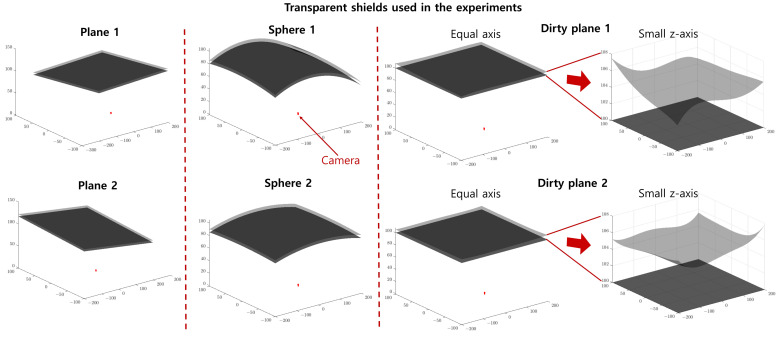
Transparent shields used in the experiments. We used three types of transparent shields: ’plane’, ’sphere’, and ’Dirty plane’ shapes. The red dot in each sub-figure represents the position of the camera. For each, we used two different shield shapes, which are listed in each row. The first and second columns include transparent shields with planar and spherical shapes. The third column provides the shield with the shape of the dirty plane. It resembles a plane but has an uneven thickness. The last column presents the enlarged view of the ’Dirty plane’ shield. The shield of the dirty plane has uneven outer surfaces that simulate the deformed outer surfaces caused by a harsh outer environment.

**Figure 8 sensors-23-08430-f008:**
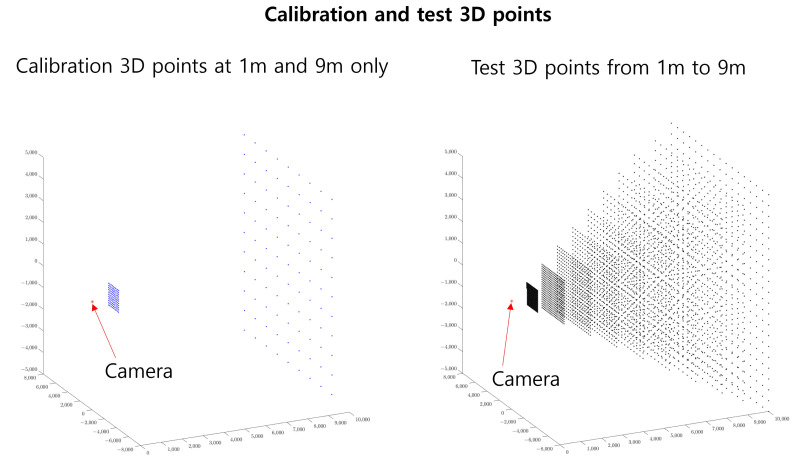
(**Left**) Calibration of 3D points where the observations exist at 1 and 9 m only for the *z*-axis. (**Right**) Test 3D points used for the experiments range from 1 to 9 m along the *z*-axis and are denser for the *x*- and *y*-axes.

**Figure 9 sensors-23-08430-f009:**
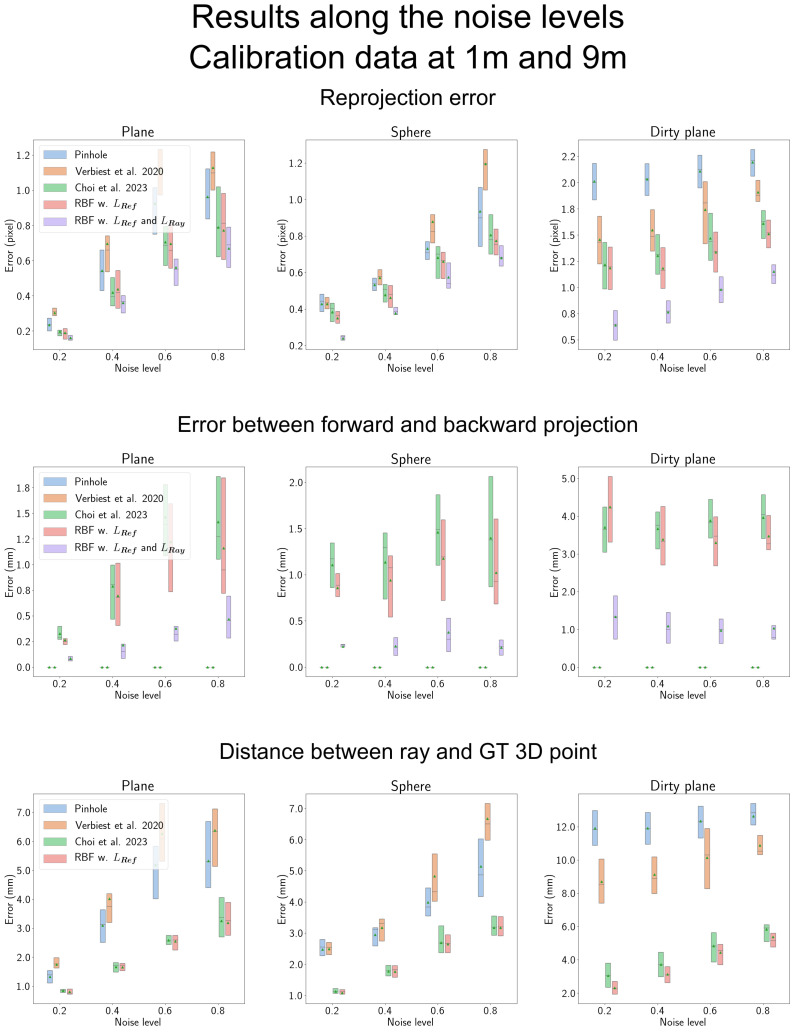
Results along the noise levels. Each column corresponds to a type of transparent shield, including plane, sphere, and ‘Dirty plane’ shields. ‘Verbiest et al. 2020’ and ‘Choi et al. 2023’ in the figure represent Verbiest et al. [13] and Choi et al. [14], respectively. In the first row, we present the reprojection error. ‘RBF w. $L_{Ref}$’ indicates the proposed RBF-based camera model optimized solely using the reference objective outlined in Equation (Equation 15). ‘RBF w. $L_{Ref}$ and $L_{Ray}$’ represents the proposed camera model optimized using both objectives as discussed in Section 3.3.3. The use of both objectives consistently yields minimal reprojection errors across all shield types. Moving to the second row, we observe a significant reduction in the error between forward and backward projection when both objectives are employed. In the last row, we evaluate the distance between the estimated ray and the ground-truth 3D point, focusing on the backward model exclusively. Therefore, we do not include the model utilizing both objectives in this analysis. Notably, ‘RBF w. $L_{Ref}$’ demonstrates superior performance compared to other methods.

**Figure 10 sensors-23-08430-f010:**
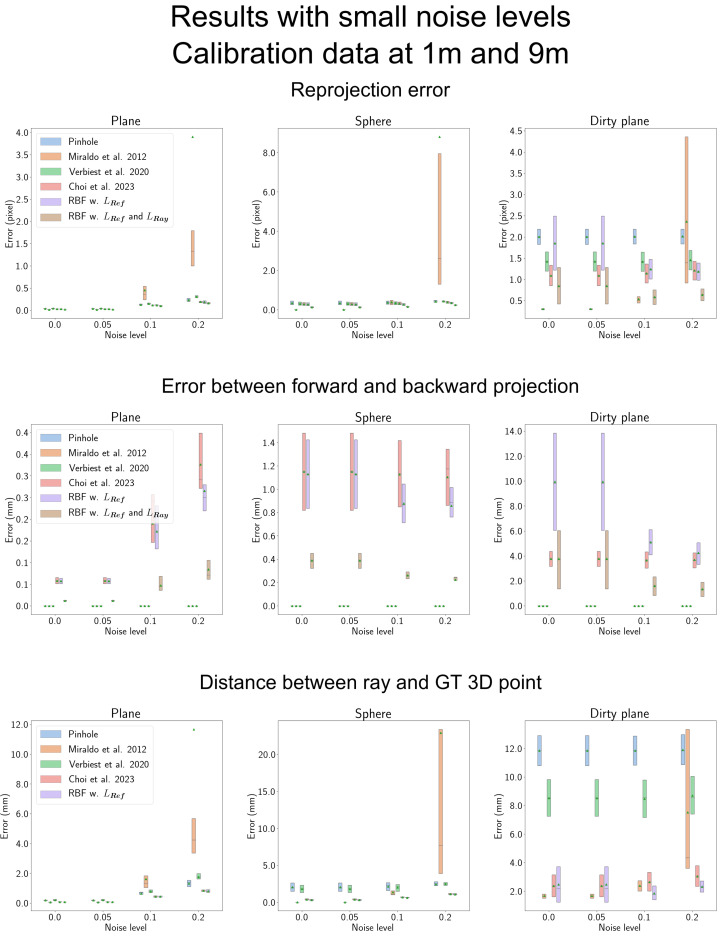
Results with a lower noise level between 0 and 0.2. ‘Miraldo et al. 2012’, ‘Verbiest et al. 2020’, and ‘Choi et al. 2023’ in the figure represent Miraldo et al. [11], Verbiest et al. [13], and Choi et al. [14], respectively. Miraldo et al. [11] reported low errors at low noise levels, but these errors increased significantly when the noise level reached 0.1 or 0.2. This phenomenon can be attributed to the optimization objective employed by Miraldo et al. [11], which lacks a constraint for the orthogonality of two vectors, namely, the moment and direction vectors in the Plücker coordinate system. As noise levels increase, the angles between these vectors deviate further from the ideal 90 degrees, as elucidated by Choi et al. [14], resulting in higher errors. With the exception of the Miraldo et al. [11] model, the proposed RBF-based camera model consistently outperforms other methods, aligning with the results presented in Figure 9.

**Figure 11 sensors-23-08430-f011:**
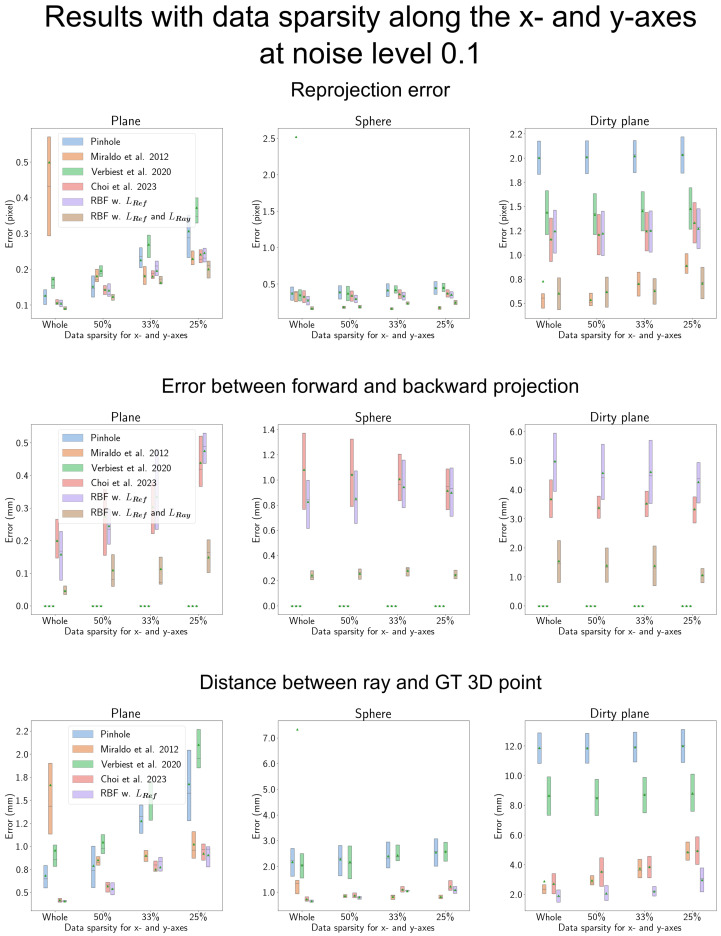
Result for the data sparsity along the *x*- and *y*-axes with low noise level. ‘Miraldo et al. 2012’, ‘Verbiest et al. 2020’, and ‘Choi et al. 2023’ in the figure represent Miraldo et al. [11], Verbiest et al. [13], and Choi et al. [14], respectively. In cases where the shield shape is simple, the performance of the proposed RBF-based camera model closely resembles that of Miraldo et al. [11]. However, when dealing with the more complex ‘Dirty plane’ shield, the proposed RBF-based camera model consistently outperforms other methods. Furthermore, even as data sparsity increases, the proposed RBF-based camera model demonstrates robustness. The reprojection error of the proposed camera model depends on the distance between the GT 3D point. Since the proposed ray constraint requires an accurate estimated ray, the accuracy of the estimated ray directly affects the reprojection error.

**Figure 12 sensors-23-08430-f012:**
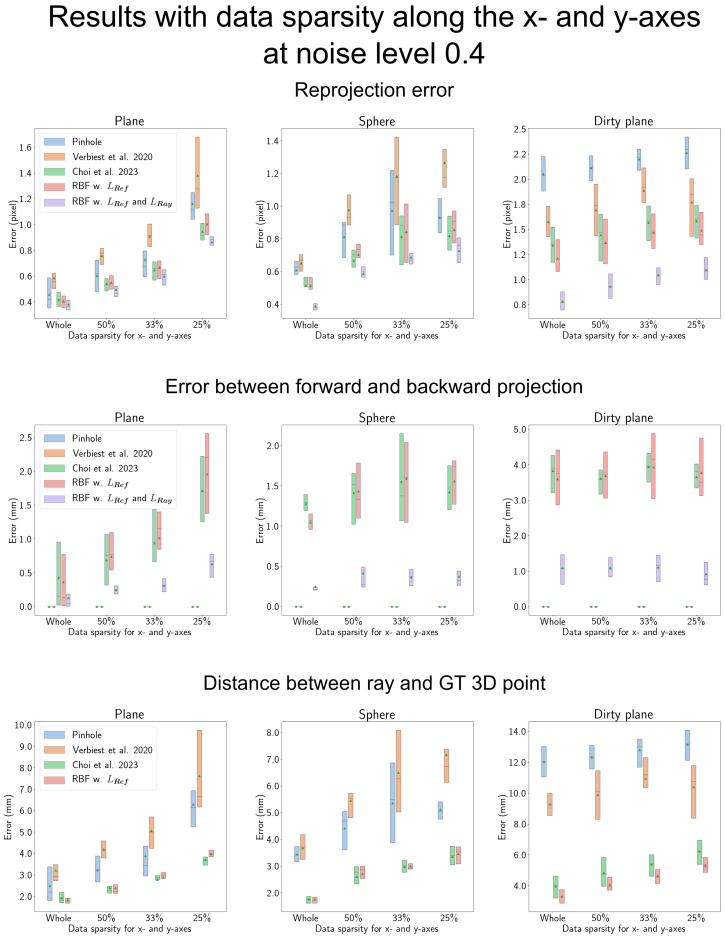
Results along data sparsity of *x*- and *y*-axes with large observation noise. ‘Verbiest et al. 2020’ and ‘Choi et al. 2023’ in the figure represent Verbiest et al. [13] and Choi et al. [14], respectively. The proposed RBF-based camera model demonstrates its effectiveness in terms of reprojection error and the error between forward and backward projections. This indicates that the proposed ray constraint remains robust even when dealing with sparse data. However, for simpler shield shapes like plane and sphere, the proposed reference constraint does not yield a significant performance improvement compared to Choi et al. [14]. In contrast, for the complex ‘Dirty plane’ shield, the proposed reference constraint consistently proves effective, irrespective of data sparsity.

**Figure 13 sensors-23-08430-f013:**
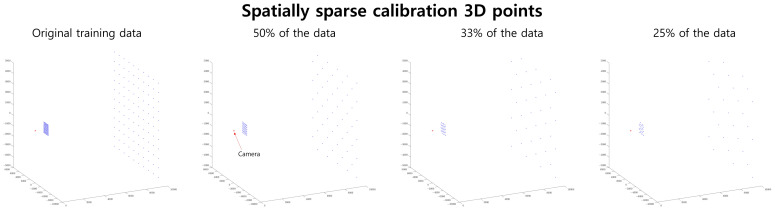
More sparse data along the *x*- and *y*-axes used in the experiment in Section 4.6. The red dot in each sub-figure represents the position of the camera. In this analysis, we deliberately omitted some of the additional 3D points to evaluate the model’s robustness to data sparsity. The calibration data are presented from left to right, showcasing the complete dataset and then with 50%, 33%, and 25% of the remaining data points.

**Figure 14 sensors-23-08430-f014:**
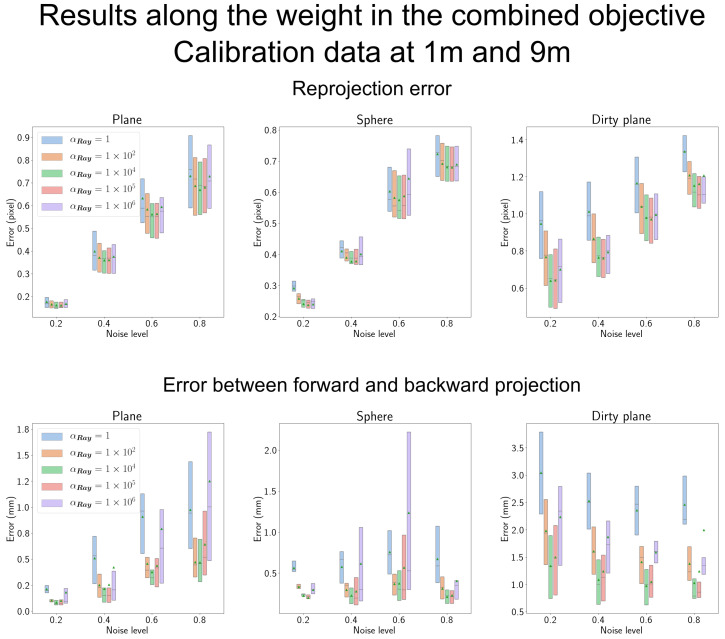
These results pertain to the $\alpha_{Ray}$ parameter discussed in Section 3.3.3. The optimal value for $\alpha_{Ray}$ is found to be $1\times 10^{4}$, indicating that the ray constraint holds greater significance than the reference objective. When the weight is increased to $1\times 10^{6}$, the performance deteriorates, suggesting that the reference objective $L_{Ref}$ prevents the residual error vectors from deviating too far from the observed errors of the pinhole camera model. We utilized $1\times 10^{4}$ for all other experiments.

**Figure 15 sensors-23-08430-f015:**
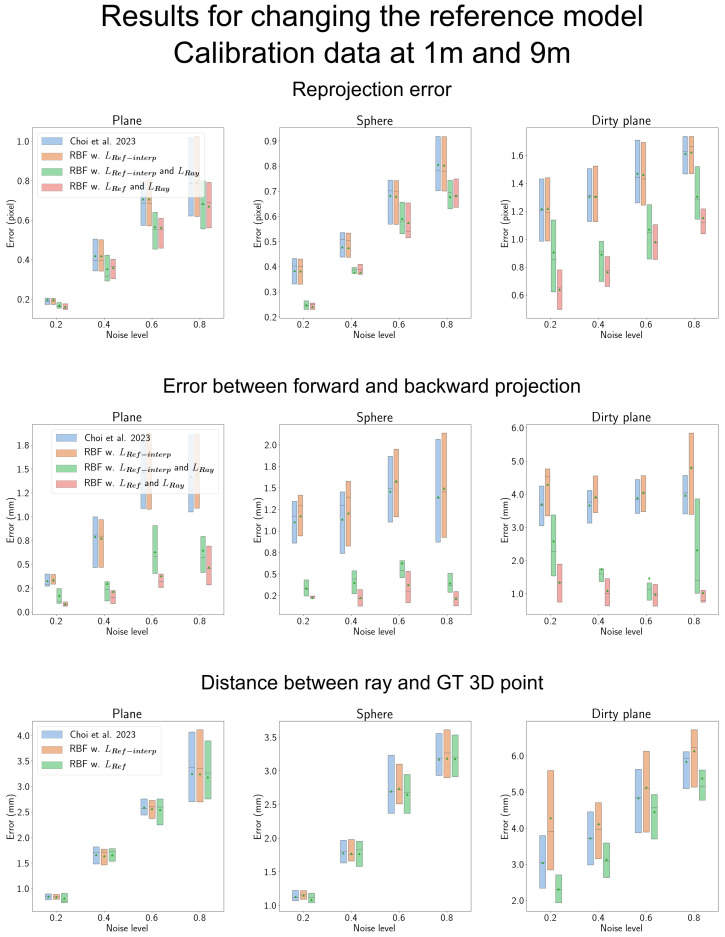
Results for changing the reference model to the interpolation method performed by Choi et al. [14]. ‘Choi et al. 2023’ in the figure represents Choi et al. [14]. We denote this as $L_{Ref-interp}$, representing the RBF regression model optimized using Choi et al.’s approach as the reference model. When we exclusively apply $L_{Ref-interp}$ without the ray constraint ($L_{Ray}$), the performance closely aligns with Choi et al.’s model [14]. However, when we incorporate $L_{Ray}$ alongside $L_{Ref-interp}$, there is a significant decrease in reprojection error and the error between the forward and backward models. Notably, for the shield of the dirty plane, employing GPR as a reference model yields superior results.

**Figure 16 sensors-23-08430-f016:**
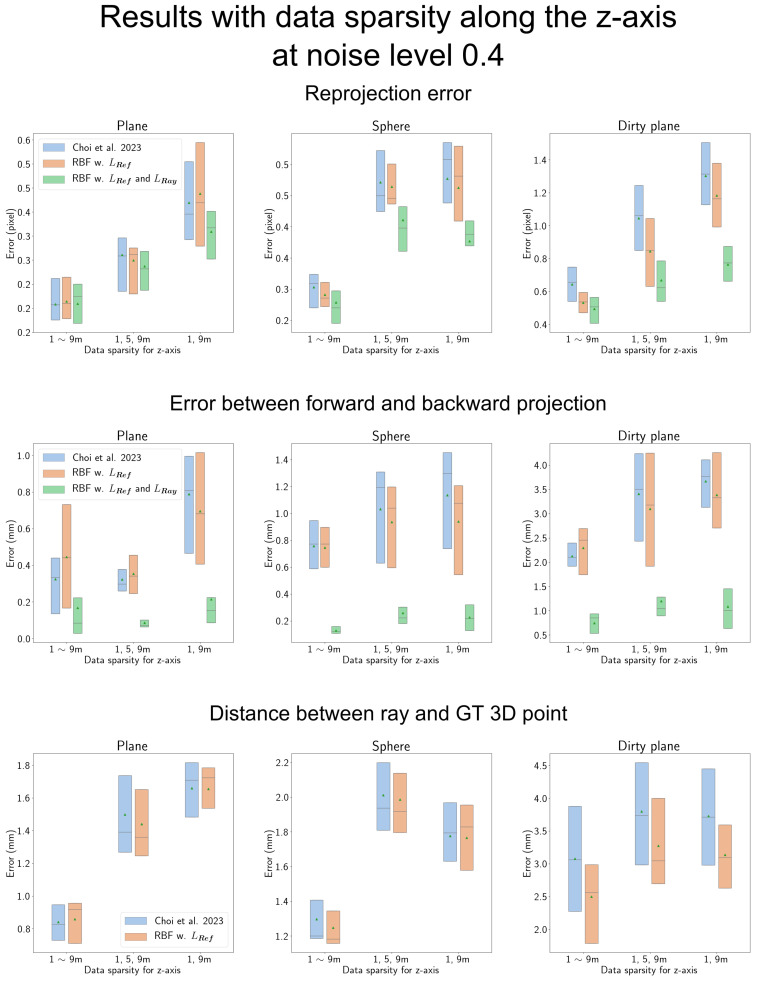
Results based on data sparsity along the *z*-axis: ‘Choi et al. 2023’ in the figure represents Choi et al. [14]. In each figure, the *x*-axis represents the level of data sparsity. As we move along the *x*-axis in each graph, the observations become sparser, ranging from 1 to 9 m, then further down to observations at 1, 5, and 9 m only, and finally at 1 and 9 m only along the *z*-axis. When the observations are dense, the ray constraint $L_{Ray}$ does not have a significant impact. However, as the data become sparser, the effectiveness of $L_{Ray}$ increases accordingly. On the other hand, $L_{Ref}$ is effective when the shape of the shield is complex.

**Figure 17 sensors-23-08430-f017:**
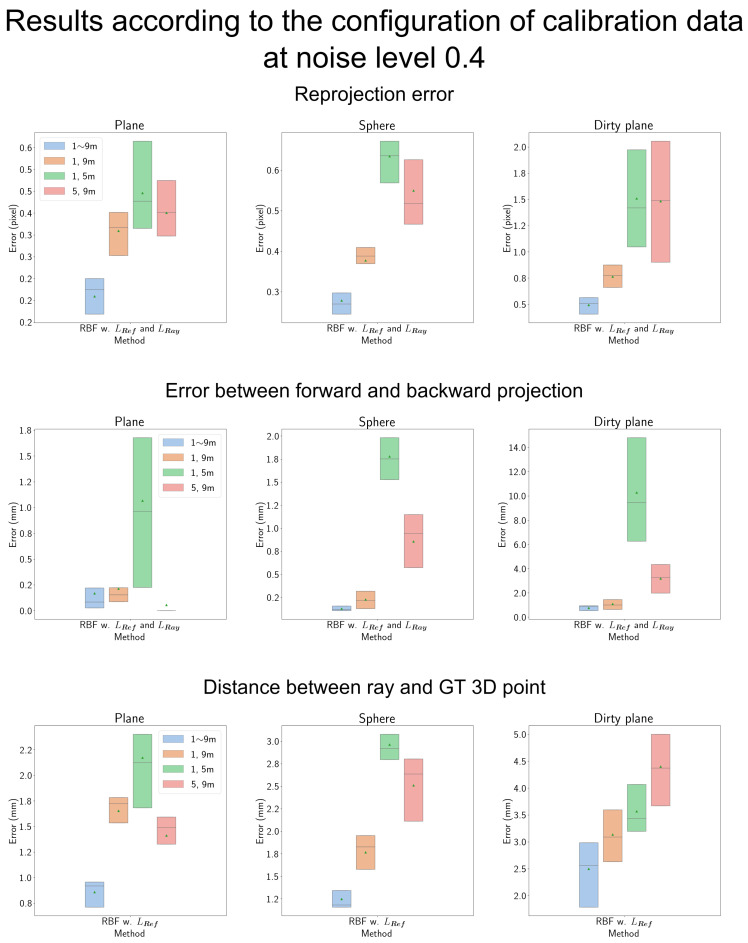
Results based on calibration data configuration: The *x*-axis represents the configuration of the calibration data, including observations spanning from 1 to 9 m, 1 and 9 m only, 1 and 5 m only, and 5 and 9 m only along the *z*-axis. In the case of the plane-shaped shield, the results for the configurations of 1 and 9 m only and 5 and 9 m only are similar. However, in all other cases, the 1 and 9 m only observation configuration consistently yields better results. This suggests that the ray constraint is more effective when the unobserved region (2 to 8 m) is situated in the well-observed region (1 and 9 m). This result is also attributed to the accuracy of the back-projected ray, which is more precise when the observation is at 1 and 9 m, especially when the shield is complex.

**Table 1 sensors-23-08430-t001:** List of abbreviations and their explanations.

Abbreviation	Full Version of the Phrase
RBF	Radial Basis Function
GPR	Gaussian Process Regression
GT	Ground Truth
3D	Three-dimensional
2D	Two-dimensional

**Table 3 sensors-23-08430-t003:** The computational time (in milliseconds) for the proposed RBF-based regression model is compared to other methods in the table. Each row presents the computation time for each method, considering both forward and backward projections. We conducted 8 trials using a total of 3948 test points and provided both the mean and standard deviation of these times. While the computation time for the proposed method is slightly longer than that of Choi et al. [14], which employs a simpler interpolation technique, it is significantly faster than the computation times required by Miraldo et al. [11] and Verbiest et al. [13]. This speed advantage is because the latter two methods involve iterative procedures for either the forward or backward projections, adding substantial computational overhead.

Method	Computation Time (ms)	
	**Forward Projection**	**Backward Projection**
Miraldo et al. [11]	20,887.9 ± 471.1	96.4 ± 2.0
Verbiest et al. [13]	32.9 ± 4.0	44,355.7 ± 1060.6
Choi et al. [14]	32.3 ± 0.5	63.9 ± 0.8
Proposed	60.6 ± 2.4	120.4 ± 1.6

## Data Availability

Not applicable.

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
