# Peer review of "RBF-Based Camera Model Based on a Ray Constraint to Compensate for Refraction Error"

_sensors, 2023, doi:10.3390/s23208430_

Round 1

Reviewer 1 Report

A camera equipped with a transparent shield can be modeled using the pinhole camera  model and residual error vectors defined by the difference between the estimated ray from the pinhole camera model and the actual three-dimensional (3D) point. To calculate the residual error vectors,  the authors employ sparse calibration data consisting of 3D points and their corresponding 2D points on the image. However, the observation noise and sparsity of the 3D calibration points pose challenges in determining the residual error vectors. To address this, the authors first fit Gaussian process regression (GPR)  operating robustly against data noise to the observed residual error vectors from the sparse calibration data to obtain dense residual error vectors. Subsequently, to improve performance in unobserved  areas due to data sparsity, the authors used an additional constraint: the 3D points on the estimated ray should be projected to one 2D image point, called the ray constraint. Finally, the authors optimize the radial basis function (RBF)-based regression model to reduce the residual error vector differences with GPR at the  predetermined dense set of 3D points while reflecting the ray constraint. The proposed RBF-based  camera model reduces the error of the estimated rays by 6% on average and the reprojection error by  26% on average.  Generally, this is a good work. It can be accepted if the authors can consider the following issues: 1. The motivations and original academic contributions should be well organized in the introduction part. 2. What is the computational load of the proposed method? 3. The NN method has a lot of applications on the estimation. More applications are welcome to enrich the literature review such as Simultaneous Robust State and Sensor Fault Estimation of Autonomous Vehicle via Synthesized Design of Dynamic and Learning Observers;Actuator Fault Reconstruction for Quadrotors Using Deep Learning-based Proportional Multiple-Integral Observer. 4. More tests are welcome to show the advantages of the proposed method.

A camera equipped with a transparent shield can be modeled using the pinhole camera  model and residual error vectors defined by the difference between the estimated ray from the pinhole camera model and the actual three-dimensional (3D) point. To calculate the residual error vectors,  the authors employ sparse calibration data consisting of 3D points and their corresponding 2D points on the image. However, the observation noise and sparsity of the 3D calibration points pose challenges in determining the residual error vectors. To address this, the authors first fit Gaussian process regression (GPR)  operating robustly against data noise to the observed residual error vectors from the sparse calibration data to obtain dense residual error vectors. Subsequently, to improve performance in unobserved  areas due to data sparsity, the authors used an additional constraint: the 3D points on the estimated ray should be projected to one 2D image point, called the ray constraint. Finally, the authors optimize the radial basis function (RBF)-based regression model to reduce the residual error vector differences with GPR at the  predetermined dense set of 3D points while reflecting the ray constraint. The proposed RBF-based  camera model reduces the error of the estimated rays by 6% on average and the reprojection error by  26% on average.  Generally, this is a good work. It can be accepted if the authors can consider the following issues: 1. The motivations and original academic contributions should be well organized in the introduction part. 2. What is the computational load of the proposed method? 3. The NN method has a lot of applications on the estimation. More applications are welcome to enrich the literature review such as Simultaneous Robust State and Sensor Fault Estimation of Autonomous Vehicle via Synthesized Design of Dynamic and Learning Observers;Actuator Fault Reconstruction for Quadrotors Using Deep Learning-based Proportional Multiple-Integral Observer. 4. More tests are welcome to show the advantages of the proposed method.

Author Response

Thank you for your valuable comments. 
Please see the attachment.

Reviewer 2 Report

This manuscript sensors-2642499 proposed sparse calibration data consisting of 3D points to calculate the residual error vectors. However, the observation noise and sparsity of the 3D calibration points pose challenges in determining the residual error vectors. To address this, we first fit Gaussian process regression (GPR) operating robustly against data noise to the observed residual error vectors from the sparse calibration data to obtain dense residual error vectors. Subsequently, to improve performance in unobserved areas due to data sparsity, we use an additional constraint: the 3D points on the estimated ray should be projected to one 2D image point, called the ray constraint. Finally, we optimize the radial basis function (RBF)-based regression model to reduce the residual error vector differences with GPR at the predetermined dense set of 3D points while reflecting the ray constraint. The proposed RBF-based camera model reduces the error of the estimated rays by 6% on average and the reprojection error by 26% on average. It was a pleasure reviewing this work and I can recommend it for publication in Sensors after a major revision. I respectfully refer the authors to my comments below.

1.         The English needs to be revised throughout. The authors should pay attention to the spelling and grammar throughout this work. I would only respectfully recommend that the authors perform this revision or seek the help of someone who can aid the authors. For example, the equation (16) is not corrected.

2.         (Section 1 Introduction) The reviewer hopes the introduction section in this paper can introduce more studies in recent years. The reviewer suggests authors don't list a lot of related tasks directly. It is better to select some representative and related literature or models to introduce with certain logic. For example, the latter model is an improvement on one aspect of the former model.

3.         Experimental pictures or tables should be described and the results should be analyzed in the picture description so that readers can clearly know the meaning without looking at the body.

4.         (Section I, Introduction) The reviewer suggest to revise the original statement as “Cameras are extensively used across diverse industries [1], including in autonomous driving.” ([1] "EHPE: Skeleton Cues-based Gaussian Coordinate Encoding for Efficient Human Pose Estimation," IEEE Transactions on Multimedia, vol. DOI: 10.1109/TMM.2022.3197364, pp. 1-12, 2023., [2] TransIFC: Invariant Cues-aware Feature Concentration Learning for Efficient Fine-grained Bird Image Classification," IEEE Transactions on Multimedia, vol. DOI: 10.1109/TMM.2023.3238548, pp. 1-14, 2023.)

5.         (Tables 3-5) All the values in this table should be with same data accuracy. The number of data after the decimal point are the same. Please check other Tables and sections.

6.         The authors are suggested to add some experiments with the methods proposed in other literatures, then compare these results with yours, rather than just comparing the methods proposed by yourself on different models.

7.         Discuss the pros and cons of the proposed RBF-based camera model.

My overall impression of this manuscript is that it is in general well-organized. The work seems interesting and the technical contributions are solid. I would like to check the revised manuscript again.

 The English needs to be revised throughout. The authors should pay attention to the spelling and grammar throughout this work. I would only respectfully recommend that the authors perform this revision or seek the help of someone who can aid the authors. For example, the equation (16) is not corrected.

Author Response

(The authors gave the same response as above.)

Reviewer 3 Report

The authors proposed an RBF-based camera model that can compensate for the refraction error caused by various transparent shield types. After a thorough reading, I am summarizing my review:

1. The overall paper is organized well, technically the paper delivers novelty.

2. Ablation study and experiments were carried out in a proper manner to validate the contributions claimed by the authors.

3. There are few structural aspects that are to be fixed in the manuscript for a better reach: 

(i). In the Related Work section, include a Table mentioning all the relevant study details with key aspects, pros and cons, parameters considered (intrinsic and extrinsic camera parameters), benchmark error rate etc. This will provide a better bird-eye view on the topic to the readers.

(ii). The limitations discussed in the conclusions can be moved and made into "Limitations and Future Work" Section just before the Conclusions section.

4. Relevant References must be included in the manuscript

Zhang, Z., 2004. Camera calibration with one-dimensional objects. IEEE transactions on pattern analysis and machine intelligence26(7), pp.892-899.

Kakani, V., Kim, H., Lee, J., Ryu, C. and Kumbham, M., 2020. Automatic distortion rectification of wide-angle images using outlier refinement for streamlining vision tasks. Sensors20(3), p.894.

5. Finally, add the list of abbreviations at the end of he manuscript for easy recall of the acronyms used in the manuscript.

Overall, I lean towards a Minor revision.

We recommend the authors to scrutinize the English sentences articulated in the Sections 3 and Conclusion. 

Author Response

(The authors gave the same response as above.)
